# Research on Wind Turbine Blade Damage Fault Diagnosis Based on GH Bladed

**Zhitai Xing [1], Yan Jia [1], Lei Zhang [2],\*, Xiaowen Song [3], Yanfeng Zhang [3], Jianxin Wu [3], Zekun Wang [4], Jicai Guo [3] and Qingan Li [4],\***

[1] College of Energy and Power Engineering, Inner Mongolia University of Technology, Hohhot 010051, China; 20211100194@imut.edu.cn (Z.X.); jia-yan@imut.edu.cn (Y.J.)
[2] Xinjiang Goldwind Science & Technology Co., Ltd., Urumqi 830000, China
[3] College of Mechanical Engineering, Inner Mongolia University of Technology, Hohhot 010051, China; sxw_istgut@imut.edu.cn (X.S.); zhyf_2014@126.com (Y.Z.); wujx@imut.edu.cn (J.W.); 20211800061@imut.edu.cn (J.G.)
[4] Institute of Engineering Thermophysics, Chinese Academy of Sciences, Beijing 100190, China; wangzk0221@163.com
\* Correspondence: zhanglei51771@goldwind.com (L.Z.); liqingan@iet.cn (Q.L.)

**Abstract:** With the increasing installed capacity of wind turbines, ensuring the safe operation of wind turbines is of great significance. However, the failure of wind turbines is still a severe problem, especially as blade damage can cause serious harm. To detect blade damage in time and prevent the accumulation of microdamage of blades evolving into severe injury, a damage dataset based on GH Bladed simulation of blade damage is proposed. Then, based on the wavelet packet analysis theory method, the MATLAB software can automatically analyze and extract the energy characteristics of the signal to identify the damage. Finally, the GH Bladed simulation software and MATLAB software are combined for fault diagnosis analysis. The results show that the proposed method based on GH Bladed to simulate blade damage and wavelet packet analysis can extract damage characteristics and identify single-unit damage, multiple-unit damage, and different degrees of damage. This method can quickly and effectively judge the damage to wind turbine blades; it provides a basis for further research on wind turbine blade damage fault diagnosis.

**Keywords:** wind turbine blade; blade damage; GH Bladed; wavelet packet analysis

## 1. Introduction

In recent years, with the continuous development of the social economy and the increasing energy demand, the consumption of fossil energy such as coal, oil, and natural gas has rapidly caused an energy shortage. New renewable energy can alleviate the energy crisis, reduce carbon emissions, and improve the social environment [1,2]. Wind energy is one of the leading renewable energy sources. The primary form of wind energy utilization is captured by wind turbines to generate electricity [3]. The global installed capacity is increasing, and the ensuing problems are also highlighted. Among them, wind turbine accidents have gradually attracted people's attention. Blade damage is one of the common fault types of wind turbines [4]. Most of the blades work in a relatively harsh environment. The blades bear complex environmental wind loads and the unit's loads. When the wind load changes suddenly and violently, it is likely to cause the wind turbine blades to flap, shimmy, and offset [5]. Blade manufacturing accounts for about 20% of the total cost [6]. If the blade is damaged, it will not only cause the unit to stop, affecting the power generation efficiency, but also increase the operation and maintenance cost of the wind farm. Therefore, the study of wind turbine blade damage fault has become one of the research hotspots of scholars at home and abroad.

At present, the research methods for wind turbine blade damage mainly focus on installing sensors on the blade, monitoring and data acquisition (SCADA) system, and nondestructive testing technology. Lyu et al. [7] reviewed the research progress of optical fiber sensors in wind turbine blade fault detection. Optical fiber sensors have good monitoring effect. However, when monitoring the running state of the blade through the sensor, the reliability of the sensor and the cost of installation and maintenance need to be considered. With the continuous development and improvement of artificial intelligence algorithms and communication technologies, SCADA-based systems have been widely used in wind farms to monitor the operating status of blades. Based on SCADA data, Wang et al. [8] proposed a fault early warning method for wind turbine blade fracture damage monitoring by constructing a deep autoencoder (DA), which can effectively identify the fracture of the early warning blade. Although the SCADA system can monitor the operation status of the entire wind turbine, it needs to deeply mine the data and sometimes rely on experience to judge the fault. Nondestructive testing technology can effectively find and identify damage, mainly based on strain measurement, acoustic emission, ultrasound, vibration, thermal imaging, machine vision, and other methods. Du et al. [9] reviewed the detection principle of nondestructive testing technology, their respective advantages and disadvantages, and future research directions. Taking pictures by UAV belongs to a kind of machine vision method; LM et al. [10] used wind turbine blade pictures to study an efficient and accurate wind turbine blade damage detection and fault diagnosis monitoring method. The results show that this method can improve the health monitoring system of wind turbine blades and improve fault detection accuracy. Similarly, to quickly and efficiently identify and detect damaged image data from wind turbine blades, Guo et al. [11] proposed a hierarchical recognition framework for wind turbine blades. The framework consists of Haar and AdaBoost algorithm and convolutional neural network (CNN) classifiers. The results show that the method has good robustness and recognition speed. Although the wind turbine blade image based on UAV can effectively detect the blade surface damage, this method still requires a certain cost and consumes manpower and material resources. Blade damage simulation based on software simulation can quickly, efficiently, and cost-effectively diagnose blade faults. Aiming at the problem of wind turbine blade pitch angle installation fault, Wang et al. [12] simulated a single blade of a 5 MW wind turbine on land through GH Bladed software and proposed a fault identification method of wind turbine blade pitch angle installation. This method can accurately identify wind turbine blade pitch angle installation fault. Feng et al. [13] proposed a new method to detect blade damage using wavelet packet energy spectrum analysis and working modal analysis. The modal parameters of the blade section are obtained using the working modal analysis method, and the damage is located according to the modal strain energy change ratio (MSECR). Finally, the results are verified by the online fault diagnosis platform integrated with MATLAB. The results show that the proposed method can accurately diagnose and locate the damage. In order to study the change law of structural frequency caused by blade damage, Gu et al. [14] studied it from two aspects of finite unit simulation and experimentation. The results show that the natural frequency of the blade is affected by the damage location, damage size, and blade speed.

The most effective and commonly used method for wind turbine blade damage fault diagnosis in actual operation is still manual observation. Therefore, a reliable and accurate online damage diagnosis method is urgently needed. In this study, GH Bladed software is used to simulate blade damage. In Section 2, the main performance parameters of the wind turbine and the important parameter settings of GH Bladed are introduced. Unlike SCADA systems, GH Bladed is a reliable wind turbine simulation software that can simulate blade faults and obtain fault information without requiring much data. Then, the fault simulation process is introduced in Section 3. In Section 4, the characteristic parameter energy is extracted by a wavelet packet for damage identification. Finally, blade damage identification detection is realized. Although this study can quickly and accurately identify and diagnose whether the wind turbine blade is damaged, this study still has certain

limitations. There is a certain difference between the simulated blade damage based on GH Bladed software and the real blade damage. In addition, the applicability of the method of joint fault diagnosis using GH Bladed simulation software and MATLAB software still needs to be widely verified.

## 2. Calculation Method

### 2.1. Overview of Wind Turbines

The wind turbine blade is one of the critical components of the wind turbine. The blade captures the wind energy to convert the kinetic energy into mechanical energy, which drives the generator to generate electricity. This paper uses the horizontal axis wind turbine, the rated power as $P = 2$ MW, the wind wheel is three blades upwind, the diameter as $D = 80$ m, the hub height as $h = 61.5$ m, and the unit level IIA. The cut-in wind speed of the unit is $V_{in} = 4$ m/s, the rated wind speed $V_r = 12$ m/s, and the cut-out wind speed $V_{out} = 25$ m/s. This paper uses the 2 MW demo wind turbine model file for research. The main parameters of the wind turbine are shown in Table 1.

**Table 1.** Main performance parameters of 2 MW wind turbine.

| Parameter | Value | Type | Parameter |
|---|---|---|---|
| nominal power (MW) | 2 | cut-in wind speed (m/s) | 4 |
| impeller direction | upper drift | cut-out wind speed (m/s) | 25 |
| number of blades | 3 | rated speed (rpm) | 12 |
| impeller diameter (m) | 80 | power control mode | variable speed pitch |
| hub height (m) | 61.5 | motor mode | variable speed generator |

### 2.2. Wind Turbine Grade

The design standard of the wind turbines in this paper adopts IEC61400-1 [15,16], in which the wind turbine grade is IIA, and the turbulence expectation $I_{ref}$ corresponding to IEC61400-1 is 0.16. The basic parameters of the wind turbines are shown in Table 2.

**Table 2.** Grade setting of wind turbine.

| Wind Turbine Grade | I | II | III | S |
|---|---|---|---|---|
| $V_{ref}$ (m/s) | 50 | 42.5 | 37.5 | |
| A $I_{ref}$ | 0.16 | 0.16 | 0.16 | Parameters are specified by the designer |
| B $I_{ref}$ | 0.14 | 0.14 | 0.14 | |
| C $I_{ref}$ | 0.12 | 0.12 | 0.12 | |

### 2.3. Calculation Software

The load calculation software used in this paper is GH Bladed 4.3, a comprehensive software developed by GL Garrad Hassan for wind turbine performance and load calculation [17]. GH Bladed can not only simulate the operation state of different types of wind turbines, but also simulate various faults of wind turbines. In addition, GH Bladed can calculate and simulate various performances and loads of wind turbines and has been widely used in engineering practice. In GH Bladed, parameters such as wind turbine blades, blade airfoils, impellers, towers, generators, nacelles, control systems, and wind models can be set.

## 2.4. Wind Speed Simulation

The operating state of this paper occurs under random wind excitation, so the wind field model is characterized by random wind load acting on the wind turbine blades, and the wind speed needs to be simulated first. When the wind turbine is in the operating environment, the wind field model acting around the blade is characterized by instantaneous wind speed. By constraining the X direction along the wind direction, the Z direction perpendicular to the wind direction and upward along the axis of the wind turbine tower, the three-dimensional coordinates of the right-handed Y direction perpendicular to X and Z, the wind spectrum type is selected as the Kaimal model, the 3D turbulent wind file is then generated according to the IEC 61400 standard.

## 2.5. Wind Turbine Blade Parameter Setting

The blade model is the primary data of software simulation calculation. The blade data directly affects the simulation calculation results, so the model must be accurate. Blade data is generally provided by the blade factory, including blade distance from the blade root position, chord length, twist angle, relative thickness, pitch axis position, and airfoil section, here, only the airfoil section number was selected, specific airfoil data is in the 'Aerofoil submenu entry'.

In the actual operation of the wind turbine, the deflection change of the wind turbine blade can obtain information through the sensor. In addition, the wind turbine's voltage, current, and generator torque are important index parameters for monitoring the regular operation of the wind turbine, and these characteristic parameters are easy to obtain during monitoring. Therefore, this paper initially selects blade deflection, wind turbine voltage, current, and generator torque as characteristic parameters for identifying blade damage. In order to eliminate factors such as instability in the initial operation of the wind turbine, the initial operation time is 7 s, the actual operation time is 600 s, the time interval is 1 s, and the total operation time is 607 s. At the same time, to clearly show the damage difference, the middle operation period is selected as the result display, in which the blade deflection is 306–310 s, and the generator torque and wind turbine current are 302–306 s.

The research method of this paper first inputs the characteristic parameters of wind turbine blades, the generated wind files, and damage information into GH Bladed software, then selects the damage characteristic parameters, calculates the damage dataset, and finally inputs the dataset into MATLAB software. The wavelet packet analysis damage dataset is used to extract energy characteristics, and the wind turbine blade damage fault diagnosis is realized; the research flow chart is shown in Figure 1.

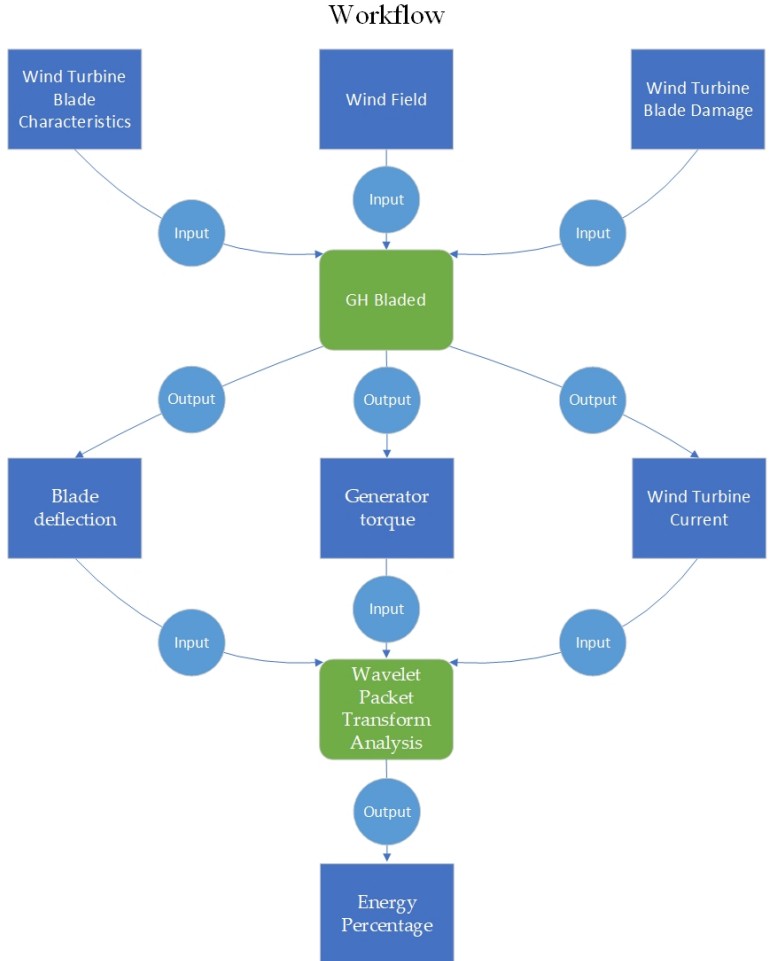

**Figure 1.** Research Flow Chart.

### 3. Results

*3.1. Simulating Blade Damage Failure*

The GH Bladed software is used for simulation, and the 2 MW wind turbine model is provided according to the software to simulate the blade damage. The blade is divided into ten units, each of which can be set separately. The research shows that when the blade is damaged by cracks and cracking, its structural stiffness will also change [18,19]. Therefore, it is feasible to simulate the blade damage by setting the stiffness parameters, and the different decrease percentage of the same unit stiffness simulates the different damage degree here. The damage degree is defined as *a*, the stiffness before and after the damage is *E* and *E\**:

$$a = \frac{E - E^*}{E} \tag{1}$$

In the blade setting module, there are mainly four setting units: blade information, blade geometry, mass and stiffness, and additional mass information that needs to be added. The blade information includes blade length, installation angle, pitch angle, blade airfoil data, and whether it is frozen. The blade geometry can be set according to the initial setting. The main parameters are length, chord, torsion angle, thickness, etc. The main parameters set in this paper are the blade mass and stiffness unit. This paper speculates two kinds of damage conditions: single-unit damage and multiple-unit blade damage. Single-unit damage simulation: the first unit is selected to simulate blade root damage, the fifth unit is used to simulate middle blade damage, and the tenth unit is used to simulate blade tip damage. Multiple-unit damage simulation: one unit, five units, and ten units are selected for simultaneous damage. According to IEC 61400-23 standard and the German

classification society wind turbine standard certification guide [20,21], slight damage is defined as a change in blade stiffness of not more than 10%, and there is no apparent visible crack or other damage traces; medium damage is defined as a change of blade stiffness between 10% and 20%, and some small cracks, wear, or deformation may appear. Severe damage and extreme damage are more than 20% of the blade stiffness change, and there are visible cracks, wear, or deformation damage marks, which may even lead to blade fracture or failure. In summary, the degree of damage in this paper divides damage into four categories: mild, moderate, severe, and extreme. Minor damage is defined as 90% of the initial stiffness, and moderate, severe, and extreme damage is defined as 80%, 60%, and 50% of the initial stiffness, respectively.

### 3.2. Analysis of Damage Characteristics

Blade deflection: in this paper, the deflection damage of a single-unit blade is simulated to obtain Figure 2, blade deflection change with time. It can be seen that the damage to the root and tip of the blade does not change significantly, and the damage to the middle part of the blade increases with the degree of damage. The deflection change is more prominent, and the offset is larger. Then, the simultaneous damage of multiple units is simulated; by simultaneously damaging the first unit, the fifth unit, and the tenth unit, and measuring the positions of the three units at the same time, the deflection change of the blade in Figure 3 changes with time, and the deflection change of the blade gradually increases with the increase of the degree of damage. In addition, the damage from multiple units has more obvious changes than from a single unit. The more significant deflection change shows that the damage of multiple units has a more significant impact on the blade, which is also in line with the actual blade damage.

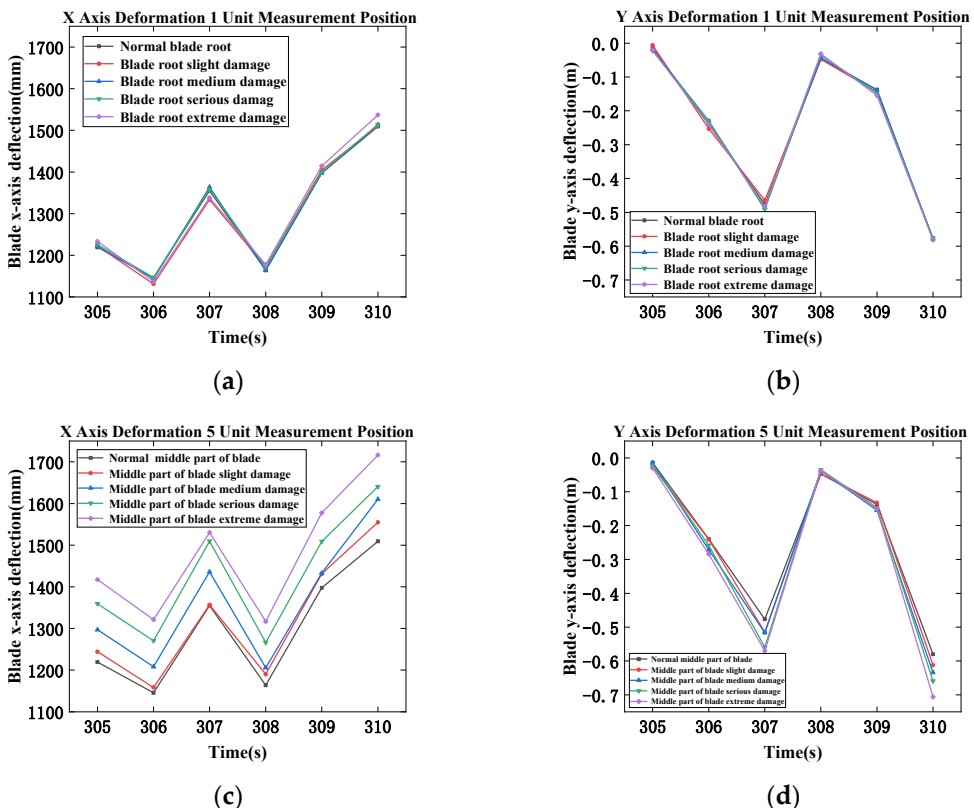

**Figure 2.** *Cont.*

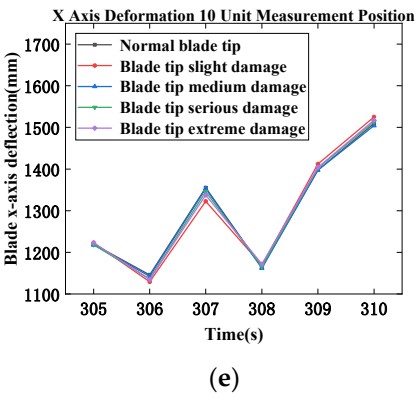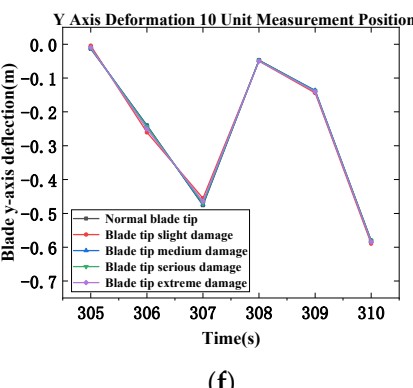

**Figure 2.** Deflection variation of *x*- and *y*-axes of blade with time: (**a**,**b**) blade root damage, (**c**,**d**) blade middle damage, (**e**,**f**) blade tip damage.

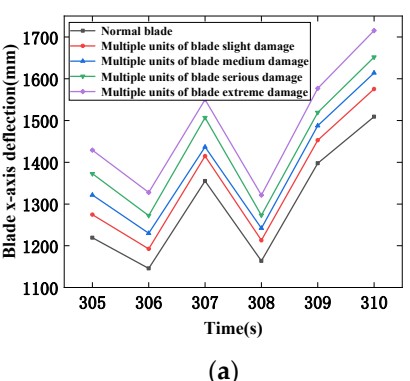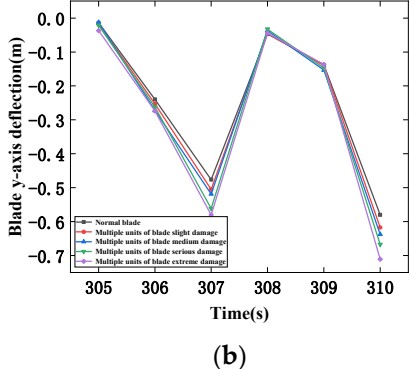

**Figure 3.** Deflection variation of multiple-unit blade damage: (**a**) *x*-axis, (**b**) *y*-axis.

Generator torque: When the blade is damaged, the wind wheel drives the transmission bearing to produce periodic vibration, which affects the generator torque [22]. As shown in Figure 4, the generator torque of single-unit and multiple-unit blade damage varies with time. It can be seen that the damage of multiple units is more evident than that of single-unit blades. In addition, overlap occurs due to the influence of generator torque fluctuation.

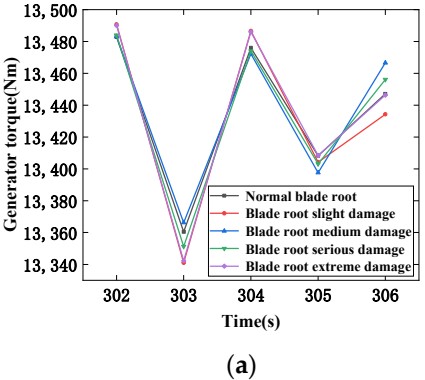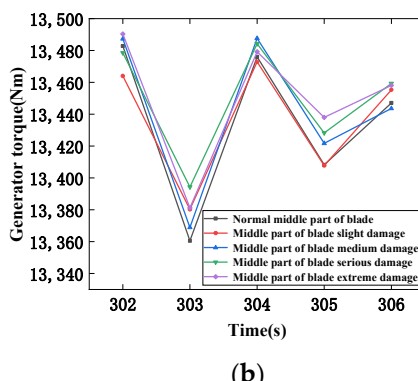

**Figure 4.** *Cont.*

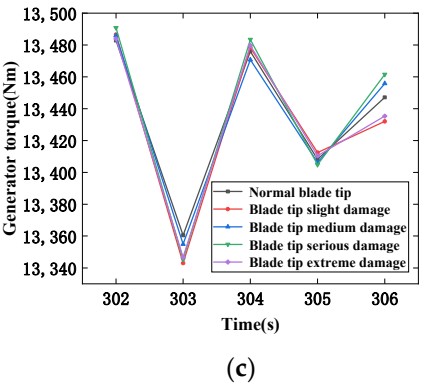
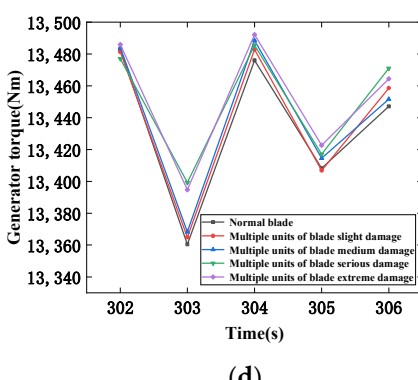

**Figure 4.** The change of generator torque with time: (**a**) blade root damage, (**b**) blade middle damage, (**c**) blade tip damage, (**d**) multiple-unit damage.

The current of the wind turbine: the periodic vibration of the generator torque will affect the current of the wind turbine. As shown in Figure 5, the current of single-blade damage and multiple-unit blade damage changes with time. It can be seen that the damage amplitude of the middle part of the blade is more evident than that of the root and tip of the blade, and the amplitude of multiple-unit damage is more significant than that of single-unit damage.

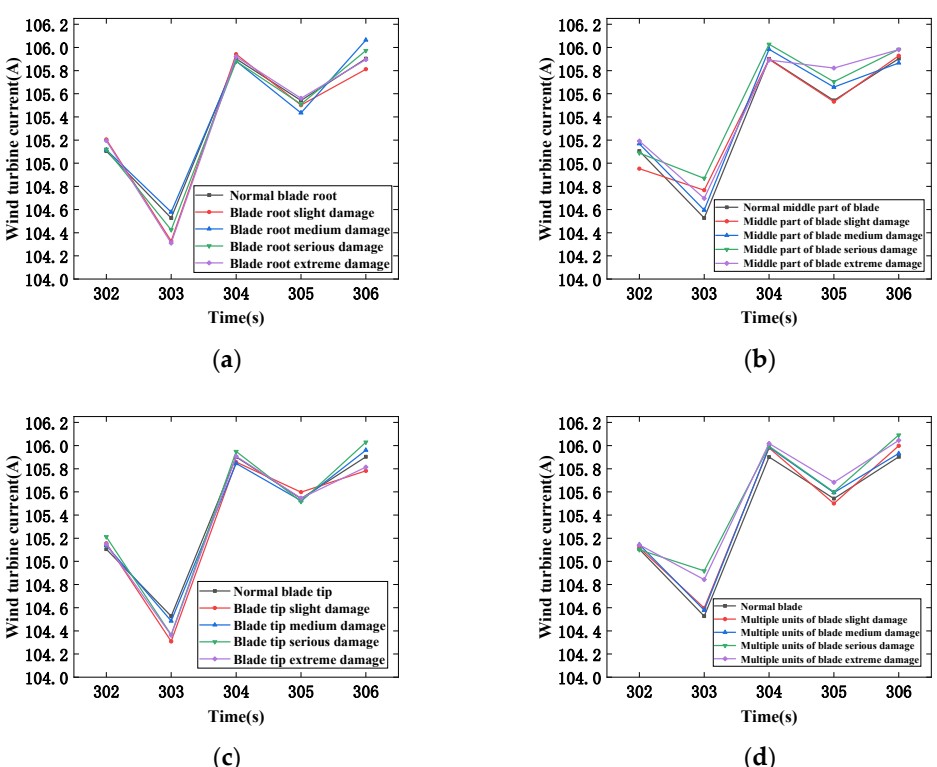

**Figure 5.** Current variation with time: (**a**) root damage, (**b**) middle damage, (**c**) tip damage, (**d**) multiple-unit damage.

## 4. Data Analysis Based on Wavelet Packet

### 4.1. Formatting of Mathematical Components

The wavelet packet transform analysis is a signal processing technology developed from the traditional orthogonal wavelet transform, which improves the deficiency that the traditional wavelet transform can only analyze the low-frequency part of the signal and cannot analyze the high-frequency band at the same time. The wavelet packet transform analysis can ensure the adaptive and translation characteristics of the window function and

distinguish the different frequency bands in the signal simultaneously. The low-frequency part and the high-frequency part can be processed. This method can effectively ensure the integrity of each detail and edge information of the signal [23,24].

After wavelet packet decomposition, the signal will get independent frequency intervals, and the energy distribution between each interval is different. Research shows that the energy distribution of each frequency interval obtained after the decomposition of the signal containing the fault is different from that of the standard signal. Therefore, the energy of each frequency interval can be extracted and compared. The specific process of extracting feature vectors is as follows:

(1) Firstly, the appropriate basis function is selected to decompose the signal $S$ into n layers, and the signal is decomposed into $2^n$ frequency bands; $i,j$ denotes the $j$ node of the n layer, $i = 1, 2, \ldots, n$; $j = 0, 1, 2, \ldots, 2n - 1$; $X_{i,j}$ represents the wavelet packet decomposition coefficient of the $j$ frequency band. The decomposition coefficients of each low-frequency and high-frequency part of the n layer are reconstructed, and $S_{i,j}$ is used to represent the reconstructed signal corresponding to each frequency band coefficient $X_{i,j}$ to ensure that it satisfies the formula. Then, the energy of each sub-band is solved. Let $E_{i,j}$ be the frequency band energy corresponding to $S_{i,j}$:

$$E_{i,j} = \int \left| S_{i,j}(t) \right|_2 dt = \sum_{j=1}^{2n-1} \left| X_{i,j} \right|^2 \tag{2}$$

(2) Finally, each sub-band can be obtained to get the vector $T$, the feature vector:

$$T = [E_{i,0} + E_{i,1} + E_{i,2}, \ldots, E_{i,2^n - 1}] \tag{3}$$

*4.2. Signal Energy Feature Extraction and Analysis*

According to the basic principle of wavelet transform and wavelet packet analysis and reconstruction, the wavelet packet function used in this paper is sym4 [25], the sampling frequency is 128 Hz, the decomposition layer is made of three layers, and the wavelet packet analysis and reconstruction model is established and programmed in the mathematical software MATLAB. The x- and y-axis deflection characteristics are selected for signal energy feature extraction, and the energy feature differences between the typical blade and single-unit damage and multiple-unit damage are compared. As shown in Figures 6–9, compared with the typical blade, the energy of single-blade damage and multiple-unit damage is concentrated in two frequency bands, which can clearly distinguish the standard blade from the damaged blade. In addition, compared with single-unit damage, multiple-unit damage energy is more concentrated in the first frequency band energy, and the energy value accounts for a more significant proportion. The results show that wavelet packet analysis can quickly identify standard blades and damaged blades, detect the occurrence of damage in a timely manner, and avoid more serious accidents caused by the accumulation and evolution of blade damage.

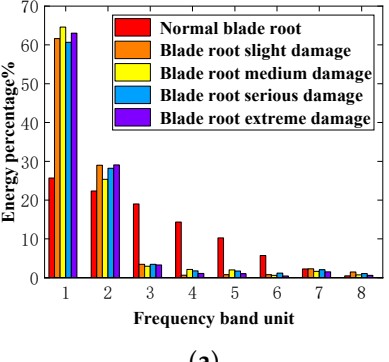

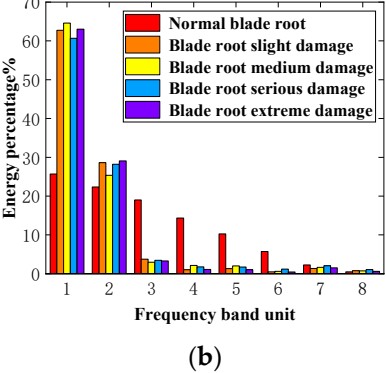

(**a**)  (**b**)

**Figure 6.** Energy feature extraction of blade root deflection damage: (**a**) *x*-axis, (**b**) *y*-axis.

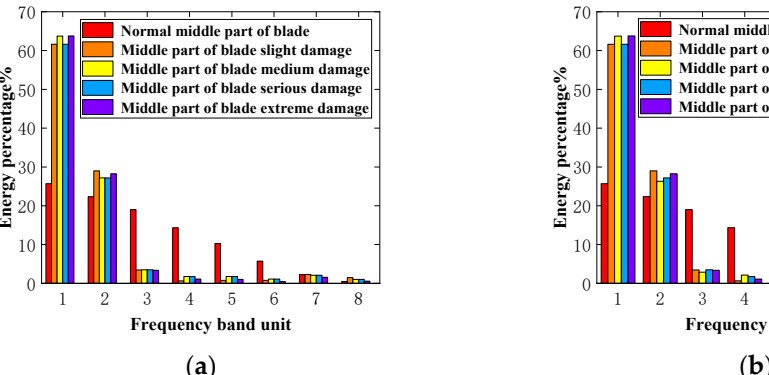

**Figure 7.** Energy feature extraction of blademiddle damage: (**a**) *x*-axis, (**b**) *y*-axis.

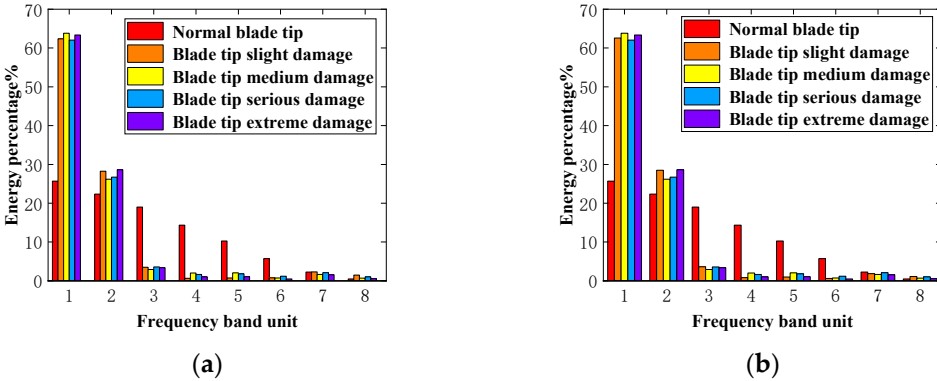

**Figure 8.** Energy feature extraction of blade tip damage: (**a**) *x*-axis, (**b**) *y*-axis.

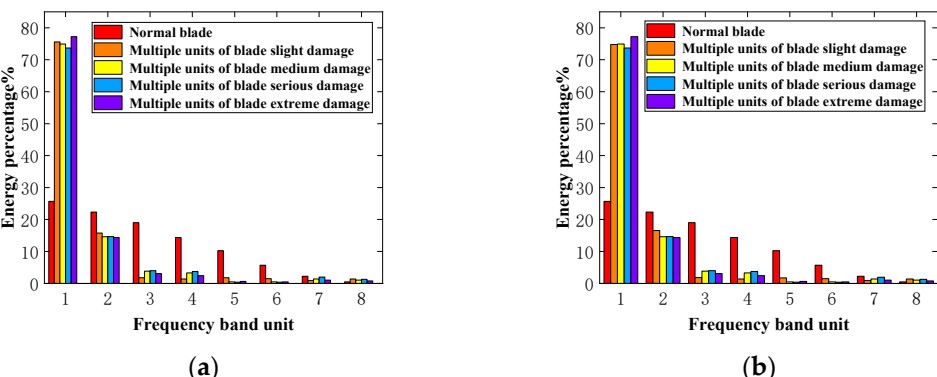

**Figure 9.** Energy feature extraction of blade multiple units damage: (**a**) *x*-axis, (**b**) *y*-axis.

## 5. Conclusions

Based on GH Bladed software, this paper simulates blade damage, single and multiple-unit damage, and sets different damage degrees for comparison. GH Bladed can initially simulate unit damage in different parts and then select a specific damage. The characteristic parameters of wind turbine operation change. Finally, the signal energy characteristics are extracted by wavelet packet change.

(1) GH Bladed selected different characteristic parameters, and the damage changes of three different parts of the blade root, the middle part of the blade, and the tip were compared. The damage to the blade's middle part and multiple units' simultaneous damage significantly influenced the change of characteristic parameters.

(2) Based on wavelet packet analysis of signal data, the energy percentage of the typical blade is significantly smaller than that of the damaged blade in the first frequency band. In addition, the energy percentage of the first frequency band of the multiple-unit damaged

blades is higher than that of the single damaged blades, which can aid in the identification of regular blades and damaged blades, single-unit damage, and multiple-unit damage.

**Author Contributions:** Conceptualization, L.Z. and Q.L.; writing of the manuscript, Z.X. and Y.J.; updating of the text, X.S. and Y.Z.; literature searches, X.S., Z.W. and J.G.; critical reviewing of the manuscript, Z.X., X.S. and J.W.; organization and editing of the manuscript, Z.X., X.S. and Y.Z.; project administration: J.W.; supervision: Q.L. All authors have read and agreed to the published version of the manuscript.

**Funding:** This research was funded by [The financial support by National Key R&D Program of China] grant number [2022YFE02070], [The National Natural Science Foundation of China] grant number [52176212], [The Program for Innovative Research Team in Universities of Inner Mongolia Autonomous Region] grant number [NMGIRT2213], and [Inner Mongolia Science and Technology Program] grant number [2021GG0336].

**Institutional Review Board Statement:** Not applicable.

**Informed Consent Statement:** Not applicable.

**Data Availability Statement:** The data supporting the reported results cannot be shared at this time, as it has been used in producing more publications on this research.

**Acknowledgments:** The authors appreciate the support of the project fund and the recognition of experts.

**Conflicts of Interest:** The authors declare no conflict of interest.

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
