# Peer review of "Research on Wind Turbine Blade Damage Fault Diagnosis Based on GH Bladed"

_jmse, doi:10.3390/jmse11061126_

Round 1

Reviewer 1 Report

This study compares various damage levels and mimics blade damage as well as single and multiple-unit damage. The results show that the proposed method based on GH Bladed can quickly and effectively judge the damage to wind turbine blades. 

Present form of article needs improvement via mention fowling points:

1. Abstract needs improvement and expansion, it should be revised adding novelty work and physical applications and the main quantitative results that appeal to a general audience.

2. Introduction is not up to the mark. Recent related studies should be added in the introduction.

3. The structure of this article can be strengthened. At the end of the Introduction section, the authors should clarify the gap between the existing research work and the work you intend to do. A brief introduction of this article and research strategy (flow chart ) should be presented.

4. Please include the significance of this work.

5. The authors have to clearly state the limitations of their study.

English language should be revised through the paper, there are some typos/grammatical errors.

Author Response

Dear reviewer:

Thank you for your lovely comments and advice. Those comments are all valuable and very helpful for revising and improving our paper, as well as the essential guiding significance to our research. We have studied the comments carefully and made a correction that we hope meets with approval. The leading corrections in the paper and the responses to the reviewer’s comments are as follows:

Response to the reviewer's comments:

Question1:1.Abstract needs improvement and expansion, it should be revised adding novelty work and physical applications and the main quantitative results that appeal to a general audience.

Response1: We are grateful for the suggestion. We have expanded and improved the content of the abstract, with modifications highlighted in yellow font.

Question2、3、4、5:2. Introduction is not up to the mark. Recent related studies should be added in the introduction.3. The structure of this article can be strengthened. At the end of the Introduction section, the authors should clarify the gap between the existing research work and the work you intend to do. A brief introduction of this article and research strategy (flow chart) should be presented.4. Please include the significance of this work. 5. The authors have to clearly state the limitations of their study.

Response2、3、4、5: Thank you for your careful review. We have revised and added the introduction to recent relevant research, clarifying the significance of the research and the limitations of the research work at the end. Finally, a flowchart for the study of this article is provided at the end of the calculation method. The modified parts are highlighted in yellow.

We would love to thank you for allowing us to resubmit a revised copy of the manuscript, and we highly appreciate your time and consideration.

Reviewer 2 Report

In this paper, the authors study turbine blade damage using commercial software. Although the number of studies described by the authors in this paper is rather limited, the results are clearly presented and the conclusions are correctly drawn. I recommend this paper for publication as is.

Author Response

Dear reviewer:

Thank you for your recognition and encouragement of this study. I wish you good health and smooth work.

We have made some modifications and improvements to the research content, and we highly appreciate your time and consideration.

Reviewer 3 Report

This is an interesting paper which showcases the application of wavelet packet analysis to predict fault diagnosis using the results of the simulation tool. There is certainly innovation in this work, however it is this reviewer's opinion that the end goal, methods and the results are not always clearly explained, and there should be some improvement.

The following are observations according to line number and chapter:

Chapter 2.4 Win speed simulation

- line 102: a short explanation why the Kaimal model was preferred over other models (e.g. von Karman) would be desirable.

Chapter 3.1 Simulating Blade damage failure

A general observation is that this chapter (or at least portions of it) should be into a chapter about methodology. (At least to my understanding) this chapter describes how damage is introduced inside the GH Bladed software (which confess I am not a user so I don't know the extent of its capabilities). 

(Again my understanding is that ) the GH Bladed software exports the characteristic parameters for this study (blade deflection, wind turbine voltage, current, and generator torque) which are then processed by the wavelet packet methodology. 

The above is to my understanding the workflow, which I am not 100% percent certain that this is the case. So I find desired a more extensive methodology and maybe a flowchart showcasing the input and outputs at each stage. 

Chapter 3.2 Selection of Damage Characteristic Parameters

(Again,) at least the first paragraph should be in methodology. 

Another issue I am having is with the way Figures are referenced and commented in the text. Figures 1 to 3 are referenced as "Figure 1-3" and are collectively referred to. To me it would have been preferable to have one Figure with 6 subplots instead, since all of them exhibit similar trends. I would also have liked more extensive description of the parameters i.e. the y-label on figures is "Blade x-axis deformation", there is no indication where the x-axis deformation is measured and also x and y axis are not explicitly stated. 

Regarding Figure 4, I find there is something missing. In Section 3.1 multiple element damage is defined as:  "Multi-element damage simulation: one element, five elements, and ten elements are selected for simultaneous damage.", however I am only seeing one graph for varying damage level (or so I interpret it. So, there needs to be some clarification on this graph. 

Also a list of abbreviations would be helpful if included. 

Although the English language used in the paper is most of the time good, in some cases there are some instances that its hard to understand. For example:

- in line 54 and 55, "wind turbine blade pitch angle installation deviation fault" and subsequently "wind turbine blade pitch angle installation deviation fault identification method", is something that needs (at least for this reviewer explaining).  More specifically, this is an already long string of nouns and adjectives and the combined use of words "deviation" and "fault" does not make clear unless the reader refers to Wang's "Fault diagnosis of installation deviation of wind turbine blade pitch angle". So, in this case the english is not appropriate and can be improved. 

- Line 72:   Table 1 has 4 columns with headers "type", and "parameter". In my opinion, "parameter" instead of type would be better, and "value" instead of parameter.

- Line 85:  (trivial change) an additional space should be added after $I_{ref}$

- Line 174: (trivial) "multiple-unit blade damage" should be "multiple-element blade damage". The authors should keep the same names for the same concept to avoid confusion.

Author Response

Dear reviewer:

Thank you for your lovely comments and advice. Those comments are all valuable and very helpful for revising and improving our paper, as well as the essential guiding significance to our research. We have studied the comments carefully and made a correction that we hope meets with approval. The leading corrections in the paper and the responses to the reviewer’s comments are as follows:

Response to the reviewer's comments:

Question1:Chapter 2.4 Win speed simulation-line 102: a short explanation why the Kaimal model was preferred over other models (e.g.von Karman) would be desirable.

Response1: We are grateful for the question. According to the GH Bladed User Manual, the wind spectrum type can be either von Karman model or Kaimal model. However, Kaimal requires fewer parameters to be set and is easier to calculate and simulate. In addition, Kaimal is more suitable for IEC standards, while von Karman model is more suitable for ESDU standards. This study is based on IEC standards, so Kaimal model is chosen.

Question2: Chapter 3.1 Simulating Blade damage failure. A general observation is that this chapter (or at least portions of it) should be into a chapter about methodology. (At least to my understanding) this chapter describes how damage is introduced inside the GH Bladed software (which confess I am not a user so I don't know the extent of its capabilities). (Again my understanding is that) the GH Bladed software exports the characteristic parameters for this study (blade deflection, wind turbine voltage, current, and generator torque) which are then processed by the wavelet packet methodology. The above is to my understanding the workflow, which I am not 100% percent certain that this is the case. So I find desired a more extensive methodology and maybe a flowchart showcasing the input and outputs at each stage.

Response2: Thank you for your careful review and we appreciate your suggestion. Based on your suggestion, we have reedited the first paragraph of section 3.1 as part of section 2.5 methodology, and have drawn a workflow diagram, with the modified parts highlighted in green.

Question3: Chapter 3.2 Selection of Damage Characteristic Parameters (Again,) at least the first paragraph should be in methodology. Another issue I am having is with the way Figures are referenced and commented in the text. Figures 1 to 3 are referenced as "Figure 1-3" and are collectively referred to. To me it would have been preferable to have one Figure with 6 subplots instead, since all of them exhibit similar trends. I would also have liked more extensive description of the parameters i.e. the y-label on figures is "Blade x-axis deformation", there is no indication where the x-axis deformation is measured and also x and y axis are not explicitly stated. Regarding Figure 4, I find there is something missing. In Section 3.1 multiple element damage is defined as: "Multi-element damage simulation: one element, five elements, and ten elements are selected for simultaneous damage.", however I am only seeing one graph for varying damage level (or so I interpret it. So, there needs to be some clarification on this graph.

Response3: We are incredibly grateful to the reviewer for pointing out this problem. We will adjust the first paragraph of section 3.2 as part of section 2.5. Change the original Figure 1-3 to a graph with 6 subgraphs, and indicate the deformation positions of the x and y axes in the title section. In addition, a supplementary explanation has been provided regarding the original Figure 4, where multiple units were damaged at the same time as the 1st, 5th, and 10th units, and the deflection changes caused by joint detection were detected. The modified parts are highlighted in green.

Question4: Although the English language used in the paper is most of the time good, in some cases there are some instances that its hard to understand. For example:

in line 54 and 55, "wind turbine blade pitch angle installation deviation fault" and subsequently "wind turbine blade pitch angle installation deviation fault identification method", is something that needs (at least for this reviewer explaining).  More specifically, this is an already long string of nouns and adjectives and the combined use of words "deviation" and "fault" does not make clear unless the reader refers to Wang's "Fault diagnosis of installation deviation of wind turbine blade pitch angle". So, in this case the English is not appropriate and can be improved. Line 72:   Table 1 has 4 columns with headers "type", and "parameter". In my opinion, "parameter" instead of type would be better, and "value" instead of parameter. Line 85: (trivial change) an additional space should be added after $I_{ref}$. Line 174: (trivial) "multiple-unit blade damage" should be "multiple-element blade damage". The authors should keep the same names for the same concept to avoid confusion.

Response4: Thank you for your lovely comments and advice. We have removed the deviation in response to the ambiguity between "deviation" and "fault" in the "wind turbine blade pitch angle installation deviation fault" to make it easier to understand. Based on your suggestion, the title has been modified. For "multiple unit blade damage" and "multiple element blade damage", considering the applicability of the entire text, the multiple unit blade damage is used to avoid confusion, and the modified parts are highlighted in green.

We would love to thank you for allowing us to resubmit a revised copy of the manuscript, and we highly appreciate your time and consideration.
